# What Kinds of Dogs Are Used in Clinical and Experimental Research?

**DOI:** 10.3390/ani12121487

**Published:** 2022-06-08

**Authors:** Evelyn Schulte, Sebastian P. Arlt

**Affiliations:** Clinic for Animal Reproduction, Faculty of Veterinary Medicine, Freie Universität Berlin, 14195 Berlin, Germany; evelynschulte@gmx.de

**Keywords:** research dogs, Beagles, breed, evidence-based medicine, literature quality, trial dogs

## Abstract

**Simple Summary:**

The objective of this study was to evaluate the signalment of dogs used in veterinary research in six different specialties. In total, 150 randomly chosen clinical studies (25 studies per specialty) published between 2007 and 2019 were evaluated for the breed, sex, neuter status, age, and weight information of the dogs used. Breed information was given for 5.7% of the included animals. Beagles were used 1.9% of the time, which was a less significant role in research than we expected. Information about the sex of the dogs was lacking for 16.2% of the included animals, while age and weight information were missing for 22.7 and 32.7%, respectively. The neuter status was not given in 38.7% of the clinical studies. The results show deficits in the reporting of demographic data for the dogs. The need for an improvement in the documentation and/or reporting of animal signalment is obvious and should be addressed by authors, reviewers, and journal editors in the future.

**Abstract:**

Background: Dogs are widely used in research to answer questions about canine or human conditions. For the latter, research dogs are often used as models, since they are physiologically more similar to humans than other species used in research and they share similar environmental conditions. From a veterinary perspective, research findings are widely based on academic research, and thus are generated under experimental conditions. In that regard, the question arises: do the dogs used for research adequately represent the dog population seen in veterinary practice? It may, for example, be assumed that Beagle dogs are often used as experimental animals. The objective of this study was to evaluate the signalment of dogs used in veterinary research. Furthermore, we aimed to assess other relevant criteria regarding the validity of clinical trials in the context of six different veterinary medicine specialties: cardiology, internal medicine, neurology, orthopaedics, reproduction, and surgery. Methods: A literature search was conducted and 25 studies per specialty were randomly selected. The breed, sex, neuter status, median age, and median weight of the dogs used for clinical studies (*n* = 150) published between 2007 and 2019 were evaluated. Results: In total, 596,542 dogs were used in the 150 trials. Breed information was given for 33,835 of these dogs (5.7%). Of the latter, 1.9% were Beagles. Nine clinical trials exclusively used Beagles. The most frequently used breeds were German Shepherds (7.3%), Labrador Retrievers (6.7%), and Golden Retrievers (4.7%). The major reporting deficits found were missing breed specification in 25.3% of the articles; missing information about the sex of the dogs in 16.2%; missing age and weight information in 22.7 and 32.7%, respectively; and missing neuter status in 38.7% of the clinical studies. The median sample size was 56 (Q1:29; Q3:365) dogs. Conclusions: The presented project revealed that Beagle dogs represent only a small proportion of dogs in veterinary research. Based on the evaluated publications, it seems that some relevant dog attributes differ between the specialties. The results, however, show deficits in the reporting of demographic data for the dogs. The need for an improvement in the documentation and/or reporting of animal signalment is obvious and should be addressed by authors, reviewers, and journal editors in the future.

## 1. Introduction

In medical research, dogs have been and still are widely used as testing animals [1], although worldwide, actual and precise figures are not easy to retrieve because relatively few countries collate and publish research-animal statistics [2]. It has been estimated that 79.9 million animals, out of which 207,724 were dogs, were used worldwide for experimental or scientific purposes in 2015 [3]. This was a 36.9% increase in the equivalent estimated figure of 58.3 million animals in 2005 [2,3]. The most widespread use of experimental animals occurs in China, followed by Japan and the United States [4]. Around 800,000 laboratory animals were used in 2019 in the US, of which 7% (56,000) were dogs [5]. In the EU from 2017 to 2018, the number of animals used in research decreased by five per cent to around nine million animals, of which 0.3% (27,000) were dogs and cats [6]. The purposes of research projects on dogs include gaining basic biological knowledge; answering questions regarding human health by using dogs as models for the development of drugs, diagnostic tests, vaccines, and medical devices; and answering questions directly linked to canine physiology or diseases [7]. Dogs are often preferred as models for human conditions because they are physiologically and clinically more similar than other species such as mice [8], and pet dogs also share the environmental conditions of their owners. In addition, the domestic dog, *canis familiaris*, reportedly bears over 450 diseases; approximately 360 of these are analogous to human diseases [9]. These analogous conditions include diabetes, cancer, epilepsy, eye diseases, and autoimmune diseases, not to mention the high numbers of rare monogenetic diseases [8]. It has been stated that the most suitable and frequently utilized breed in clinical trials is the Beagle. These dogs are commonly kept in kennels in homogenous groups based on age, weight, sex, and neuter status in the research facilities of universities or pharmaceutical companies [10]. Beagles are medium-sized and have a short coat and an even temperament, which makes them particularly suitable for medical research in contrast to other breeds [11]. These advantages make it easier to standardise specific and relevant conditions in research settings and help keep the costs of the experiment lower [12].

Worldwide, more than 354 dog breeds are registered at the FCI (Federation Cynologique Internationale) [13]. Searching for “Beagle dog” in the PubMed database for results between 2007 and 2019 (http://www.pubmed.gov, accessed on 21 May 2022) leads to more than 4790 results. When performing the same search with the breed “Labrador dog”, which has been the most popular breed in the US since 1991 [14], the results are much less (1008 results). These figures support the assumption that Beagles may be the most utilized research dogs by far.

Both dog breeds can be found in the top ten list of the most popular dog breeds in the US [15] (Table 1). Beagles often serve as models for research focusing on human health, such as the toxicology of medications [16]. Research on Beagles has led to various relevant findings for human medicine, such as information about infections with *Helicobacter Pylori* [7], or better insights into the effects of the frequency of tooth brushing [17].

Depending on the research question, alternatives for the use of research dogs may be the enrolment of client-owned dogs or the retrospective evaluation of medical records. Findings from these approaches might even better represent the heterogeneity of dogs seen in daily practice and better address real clinical conditions. However, depending on the research question, a high level of heterogeneity regarding breed, age, sex, weight, neuter status, housing, feeding, and other parameters in a study population may lead to a significant influence of confounders [18,19]. If potential confounders are not taken into account during the analysis and interpretation of research results, this may lead to biased outcomes and conclusions. There is evidence that the sex, weight, age, breed, and neuter status of the dogs that take part in a clinical trial are important when it comes to different conditions and diseases, such as joint disorders [20,21], metabolic conditions [22], and periodontal diseases [23]. In addition, breed differences significantly affect the incidences of specific diseases such as pyometra [24], dilatative cardiomyopathy [25], and granulomatous colitis [26], as well as sex-steroid-influenced diseases such as diabetes [27] and hyperadrenocorticism [28].

The objective of this study was, therefore, to evaluate what kinds of dogs were utilized in clinical trials. We aimed to assess and compare information about the dogs, including the breed, sex, age, weight, and neuter status in the context of six different veterinary medicine specialties. In addition, we aimed to evaluate the extent to which Beagles have been used in veterinary research.

## 2. Material and Methods

A literature search in the databases PubMed (http://www.pubmed.gov accessed on 31 October 2020) and CAB Abstracts (https://www.cabdirect.org/, accessed on 31 October 2020) was conducted on 31 October 2020. The same literature search and selection process for articles was used and described in detail in another research project [29] assessing different literature parameters. In brief, the following search keywords were used: clinical trial AND dogs AND specialty. For each search procedure, “specialty” was replaced by cardiology, internal medicine, neurology, orthopaedics, reproduction, or surgery.

Publications had to be in the English or German language and published in or between the years 2007 and 2019. Case reports or case series with a number of animals lower than *n* < 10, opinions, clinical experiences, and abstracts with less than 500 words were excluded. Studies or case reports without statistical analysis and studies on other species, such as humans or cats, were also excluded. In addition, in vitro studies were not included. From the final 150 articles, 134 articles were accessed via online databases, nine papers were retrieved in the veterinary library at the University of Berlin, and two articles were obtained via inter-lending from other libraries. As a first step, the literature was evaluated using a slightly modified version of the checklist developed in 2010 by Arlt and Heuwieser [30]. The results have been published in a previous article [29]. In addition to the validated checklist, the following items were assessed for the presented project: number of dogs, number and type of dog breeds, gender, neuter status, median weight, and median age.

### Statistical Analysis

All statistical analyses were conducted in IBM SPSS for Windows (Version 24.0; SPSS Inc., Munich, Germany). Categorical data were presented descriptively as raw numbers and percentages. To identify differences between the specialties, the non-parametric Mann–Whitney U test was used as indicated by the distribution. The statistical significance was set at *p* < 0.05.

## 3. Results

From the 150 examined studies, 108 (72.0%) were prospective and 42 (28.0%) were retrospective. Considering the study design, 91 publications were classified as interventional studies (60.7%) and 59 were observational studies (39.3%). In total, 596,542 dogs were used in the 150 clinical studies assessed in this project. For one study, the number of dogs was not specified; instead, the number of limb fractures over a given period of time was reported. For statistical reasons, we set the number of limb fractures equal to the number of dogs. In 100 articles (66.0%), the breeds of all enrolled dogs were specified, leading to 33,835 dogs with breed information. Out of the remaining 50 trials, breed information was not given at all for 38 studies and was incomplete for 12 studies. Most studies with no or incomplete breed information were retrospective. In several studies, the breed was not specified for all dogs included, but only for the numerical top ten breeds. Analysing breed information in the 12 studies with incomplete data led to another 7792 dogs with given breeds and 5384 with missing information. Overall, breed information was available for 41,627 dogs, which was 6.9% of the overall number of dogs. The overall median number of dogs used in each of the studies was 56 (Q1:29; Q3:365). Retrospective studies had a larger number of included animals (Median: 62; Q1:35; Q3:384) than prospective ones 24 (Q1:13; Q3:41), *p* < 0.05. Out of the dogs with known breeds, 643 dogs (1.5%) were Beagles. In nine studies, the dog population consisted of Beagles only. The median sample size of these studies was 12 dogs (Q1:12; Q3:24). These studies included five experimental studies (two related to human research) and four clinical trials (one related to human research) carried out to determine the effectiveness or administration route of drugs. All studies took place under laboratory conditions. One article was published in each of the years 2007, 2008, 2014, and 2015, two articles were published in 2017, and three were published in 2018. Furthermore, in 32 studies, Beagle dogs were used among dogs belonging to other breeds. In Section 3.4, more information about the dogs used in experimental trials is given. Out of 150 studies, 16 (10.6%) were related to human research using dogs as a model, and three of these used study populations consisting of Beagles only.

Within the individual specialties, the number of Beagles used as clinical trial dogs was heterogenous (Table 2). The greatest numbers of Beagles were used in internal medicine and surgery studies (each *n* = 8, 32.0%).

For dogs with a known breed, the proportion of Beagles differed between the specialties. In internal medicine, we found the highest proportion of Beagles, with 12.9%. Orthopaedics followed with 5.6% and cardiology with 4.4%. The proportion was smallest in reproduction, with 0.4% (Table 3).

The highest amount of missing breed information was found in internal medicine (48.0%), followed by orthopaedic studies (40.0%), reproduction (36.0%), cardiology (28.0%), and neurology (20.0%). Only a portion of the dogs used in the 31 prospective studies (20.6%) and the 19 retrospective studies (12.6%) had their breeds specified. The median number of breeds was four for prospective studies (Q1:1; Q3:10), while retrospective studies had a median number of sixteen breeds (Q1:3; Q3:27). For the different specialties, the median number of breeds was one for cardiology (Q1:1; Q3:9.5), ten for internal medicine (Q1:4; Q3:15), nine for neurology (Q1:1; Q3:21), five for orthopaedics (Q1:1; Q3:16), four for reproduction (Q1:1; Q3:8.5), and nine for surgery (Q1:4; Q3:13). For 40 (26.7%) studies, there was only one breed utilized; 14 (56.0%) of these studies were cardiology trials with mostly mongrel dogs. Table 4 lists the US top ten breeds of 2019 and the number of dogs belonging to each breed that have been used in clinical veterinary research worldwide. The popularity of the dog breeds in Europe are similar [31]. These dog breeds seem to play an important role in research, since they make up around 25.2% of all study dogs with given breed information. Interestingly, despite their popularity, Bulldogs were not extensively used in the studies selected in this project.

### 3.1. Sex of the Dog Population

In total, the sex of the dogs was specified in 83.8 % (*n* = 150) of the trials (Figure 1). Dogs of both sexes were used in 89 studies (59.3%), solely females in 25 trials (16.7%), and only male dogs in 11 studies (7.3%). Both sexes were predominantly mixed in 22 neurology studies (88.0%) and 20 internal medicine studies (80.0%). In studies relating to reproduction, dogs of only one sex were used in most trials (*n* = 20, 80.0%). In seventeen studies, female dogs were used (68.0%), and in five studies, both female and male dogs were used. In contrast to the other specialties, the sex of the dogs was determined for all 25 reproduction studies. In around 40% (*n* = 10) of the studies on cardiology, the sex of the dogs was not documented, followed by six orthopaedic studies and six surgery studies with an unknown sex for the dogs used (24.0%).

**Spay and neuter status of the study population.** The spay and neuter status of most dogs used in the 150 studies was not specified in 38.7% of the studies (Figure 2). In four studies (2.7%), all dogs used in the trial were neutered. In 34.7% of the studies, both neutered and intact dogs were used, and in 24.0% of the studies, all dogs were intact. Except for studies in the field of reproduction, the neuter status was not specified as an inclusion or exclusion criterion. In 23 studies (92.0%) belonging to the reproductive field, intact dogs were enrolled (*p* < 0.05). In 18 neurology studies (72.0%) and 13 internal medicine (52.0%) studies, both neutered and non-neutered dogs were used. The spay and neuter status of the dogs used was not specified in 16 (64.0%) studies each on cardiology and orthopaedics.

### 3.2. Age of the Study Population

Overall, age information was not given in 34 (22.7%) studies (Figure 3). The median age of all dogs with given information was 5.0 years (Q1:2.4; Q3:6.9), the minimum age was 2 weeks, and the maximum age was 12.5 years. In six studies (4.0%), dogs with a median age of under one year were used. Age information was missing most often in cardiology studies (*n* = 11, 44.0%), followed by reproduction studies (*n* = 8, 32.0%). The median age for dogs enrolled into cardiology studies was 9.0 years (Q1:6.4; Q3:11.7), internal medicine was 6.0 years (Q1:4.9; Q3:7.6), neurology was 4.4 years (Q1:2.7; Q3:6.9), orthopaedics was 4.8 years (Q1:2,4; Q3:5.6) reproduction was 3.3 years (Q1:2; Q3:5.2), and surgery was 4.0 years (Q1:2.1; Q3:5.45).

### 3.3. Weight of the Dogs

Weight information was not given in 49 publications (32.7%) (Figure 4). The overall median weight was 20.3 kg (Q1:10.8; Q3:27.2), the lowest median weight of a study population was 3.7 kg, and the highest median weight of the dogs used within one study was 50.3 kg. Weight information was missing most often in reproduction studies (*n* = 17, 68.0%), followed by neurology studies (*n* = 12, 48.0%), cardiology and orthopaedics studies (each *n* = 6, 24.0%), and surgery and internal medicine (each *n* = 4, 16%). The median weight of the dogs used in cardiology studies was 14.5 kg (Q1:9.3; Q3:22.0), internal medicine was 20.2 kg (Q1:9.3; Q3:23.7), neurology was 19.3 kg (Q1:12.5; Q3:20.6), orthopaedics was 31.2 kg (Q1:25.6; Q3:33.8), reproduction was 17.8 kg (Q1:11.8; Q3:21.1), and surgery was 22.0 kg (Q1:10.7; Q3:25.2). One study in the field of surgery used dogs with an average weight of over 50 kg. The weight differences of dogs used in orthopaedics compared with neurology and cardiology were significant (*p* < 0.05).

### 3.4. Origin of Study Population and Overall Number of Animals

Information on the ownership of the dogs and the number of dogs used for the different studies has already been presented in a previous paper [29]. The dogs were either experimental animals owned by the research institutions (18.7%), privately owned (76.0%), mixed (0.7%), or the origin of the dogs was not specified (4.7%). For research in the field of cardiology, experimental animals were used in 11 of the articles (44.0 %). The dogs used for the 28 experimental studies had a median age of 2.3 (Q1:2; Q3:4.6) years and a median weight of 15.5 kg (Q1:11.9, Q3:22.5). The neuter status of the dogs was not given in twenty articles, seven study populations were not neutered, and one had a population with a mixed neuter status. For the experimental studies, the gender of the dogs was unknown for eleven study populations, only female dogs were used in five studies, only males in seven studies, and a mixed population in five studies. The median number of dogs used for experimental studies was 21.0 (Q1:11.5, Q3:24.3) and the median number of breeds was 1.0 (Q1:1.0; Q3:1.0). In 27 out of the 28 experimental studies, the breed of the dogs was specified. The study population of nine studies consisted of cross-bred dogs and seventeen study populations were purebred dogs. Most of the experimental studies (n = 9) consisted of Beagles only, followed by three studies with a mixed study population of Beagles, Labradors, Rottweilers, and German Shepherds, and one each of Foxhounds, Sheepdogs, Pitbull Terriers, and Coonhounds.

The number of animals was given in most of the articles in all different specialties. For two studies—one on reproduction and one on orthopaedics—the authors did not specify the number of enrolled animals. The median sample size for all the included studies was 31 dogs (Q1:16; Q3:64). For the different specialties, the median sample size was 32 for cardiology (Q1:22; Q3:207), 31 for internal medicine (Q1:14; Q3:53), 33 for neurology (Q1:15; Q3:56), 36 for orthopaedics (Q1:19; Q3:95), 35 for reproduction (Q1:18; Q3:74), and 25 for surgery (Q1:16; Q3:40).

## 4. Discussion

This study focused on the assessment of the signalment, such as breed, sex, neuter status, age, and weight, of dogs used in controlled clinical trials. These data were analysed with consideration of the veterinary specialty that each publication belonged to. The results of a critical appraisal of the quality of the clinical studies have been published in an earlier paper [29]. The articles were randomly selected from a list generated after a literature search in two relevant databases. There were sixteen studies using dogs that served as model for human health, but only three studies used Beagles as a study population. Interestingly, nine of these studies belonged to the specialty of cardiology.

The proportion of Beagles in our literature sample was less than we expected and less than several authors have claimed [10]. A reason could be that some studies focus on conditions that occur naturally in specific breeds, such as diseases related to brachycephaly. In 2020 in the UK, 99% of dogs used in experimental research (medical and veterinary) were Beagles [32]. In other European countries, breed information is not given in the statistics about animal testing [6]. Obviously, the Beagle has not been a predominant breed used for clinical veterinary studies in the past years. In fact, the breed of dogs used in research more or less reflects the most popular breeds, with the exception of Bulldogs. The reason for the latter remains open and should be further investigated. Nevertheless, the presented data may imply a slight increase in the use of Beagles over the last years, but the number of the assessed articles is much too small to conclude a trend. On the contrary, the number of dogs used as experimental animals seems to be declining [5], albeit a lack of comprehensive data on the worldwide use of dogs in experimental research and clinical trials. The use of Beagles and dogs in general for basic research and research related to human medicine may have become less popular in the past decades because of an increasing awareness of animal welfare [4]. Just recently, a Beagle-breeding company was inspected and found itself confronted with allegations of severe animal welfare violations, which were reported in different media [33]. Over the years, the European Union, Canada, the United States, and several other countries have introduced laws to regulate the use of laboratory animals for medical research after consulting the main stakeholders [4]. In addition, laboratory dogs are often rehomed nowadays into private households after their use in research [34,35]. However, it has been claimed that they still experience an extreme change in their life situation because they leave their familiar, limited environment in the research facility and encounter a multitude of animate and inanimate stimuli in their new home [36,37]. For 50 studies (33.3%), the amount of missing or incomplete information about the breeds used for the clinical studies is relatively high, which shows that this kind of information seems to be considered unimportant by authors, reviewers, and editors. However, since breed may be a relevant confounder, this information is essential data that needs to be given according to the STROBE statement [38]. The median number of breeds used for the clinical trials does not differ significantly between the veterinary specialties. It is noticeable that most of the cardiology studies were conducted with experimental dogs and most of the studies consisted of just one breed. The evaluation of the other parameters showed that, for most trials, both female and male dogs were used, meaning that sex was not set as an inclusion or exclusion criterion. The sex of all dogs was only given in reproduction trials, and more than 60% of these trial dogs were female. It is plausible that studies on gynaecology usually relate to conditions found in intact bitches. For the other veterinary specialties, the sex of the dogs seems to be considered unimportant, and this might indeed be the case. Similar findings have been documented regarding the neuter status. More than 90% of the dogs used for reproduction trials were intact, while in the studies belonging to the other veterinary specialties, both intact and neutered dogs were often used. It can be concluded that the neuter status seems to be considered an irrelevant factor or confounder for most research questions beyond reproduction. It is noticeable that the median age of the dogs used for cardiology was nearly double that of the ages of the dogs enrolled into studies belonging to the other veterinary specialties. This might reflect the fact that cardiac diseases tend to occur or be diagnosed in older dogs. The median age of the dogs used for the reproduction studies was the lowest. This may be related to the optimal breeding age of bitches. The median body weight of the dogs was highest in the orthopaedic studies, at around 30 kg. It is common knowledge that especially large and heavy dogs have a higher risk of suffering from joint diseases [20]. For other disorders, factors such as age and weight may also play a role concerning a studied condition. Therefore, this information should be given in scientific articles. The presented review has revealed documentation and reporting deficiencies. Based on the presented data, it is not possible to judge whether the study methodology or reporting is better or worse in specific veterinary specialties. In fact, it seems that even if some aspects are better presented in one specialty, other factors are missing more often in the same specialty. Similar shortcomings have been described by Reynolds et al. [39] and others [29,40,41]. Demographic data for dog populations used in clinical trials is very important, since it is needed to draw sound conclusions and extrapolate the findings [42]. In addition, comprehensive information about the used animals is important for readers to assess whether the given scientific information should be applied in an actual case. It has been proven that the signalment such as age, sex, and breed are highly relevant aspects for examining the prevalence of conditions and the interpretation of various study outcomes. Besides the examples mentioned earlier, age is a key factor for the outcomes of electrocardiographic exams [43], fertility [44], behaviour [45], and several canine diseases [46,47,48]. Belic et al. found that the sex of a dog plays an important role in biochemical markers for the bone turnover [49]. Anatomical or hormonal differences between male and female or neutered and intact dogs can also have an impact on the prevalence of diseases [50,51,52] or the outcome of clinical studies [53]. In addition, it has been shown that the breed has an effect on renal size [54], and genetic and phenotypic differences across dog breeds have an influence on the safety and efficacy of pharmaceutical substances and their doses [55] and the occurrence of genetic diseases [56] such as MDR1 mutations [57]. Several studies on the methodological and reporting quality of clinical trials in veterinary medicine have been published in the past years. Limitations have been criticized, such as small numbers of included animals; a lack of sufficient reporting on the specifications of animals, diagnoses, and treatments; and undocumented inclusion and exclusion criteria [29,42,58]. Our results show that relevant reporting deficiencies in clinical studies were found and essential information about the dogs used in clinical trials was missing. This may considerably limit the validity of research results. More attention should be paid to reporting guidelines, such as the STROBE statement for observational trials or the CONSORT statement for randomised studies [59], as they have been developed to improve the quality of scientific articles.

The estimated number of dogs used worldwide for medical research is still high. Attempts have been made to replace laboratory animals by in vitro and in silico methods [1,60,61,62]. For laboratory animals in most countries, the consideration and implementation of replacement, refinement, and reduction (3Rs) strategies, proposed by Russell and Burch in 1959, is mandatory [63]. For some purposes, animal testing has been forbidden by some authorities. For instance, the use of animals for cosmetic testing has been prohibited in Europe as of March 2013 [64].

### Limitations

There are some limitations to the present study, such as the relatively small sample size of 25 articles per specialty. Since this number per specialty led to a total number of 150 articles, which was eligible for a throughout assessment, the inclusion of more literature was not possible within this project. In that regard, it may be worthwhile to re-evaluate some of the presented specialty-specific findings on a larger scale with a greater number of articles.

Furthermore, the investigation and evaluation of the studies was done by just one non-blinded researcher, which may have led to a biased interpretation. This approach, however, has been used in several other studies before [29,30,41,65].

## 5. Conclusions

The results of our literature review concerning the kinds of dogs that are used in veterinary research indicate that we are widely not able to give a sufficient answer. The presented project revealed that Beagle dogs represent only a small proportion of dogs in clinical veterinary research. It seems that Beagles are used much more in experimental research, but this should be investigated in future analyses including more experimental research reports. The results of this study are furthermore in accordance with previous findings, and reflect once more that essential information about the dogs used in clinical trials is missing. Some parameters, such as body weight and neuter status, vary significantly between specialty-specific studies. Authors, reviewers, and journal editors should pay more attention to the reporting of basic information about the animals enrolled in veterinary research and should follow the guidelines for the specific study type.

## Figures and Tables

**Figure 1 animals-12-01487-f001:**
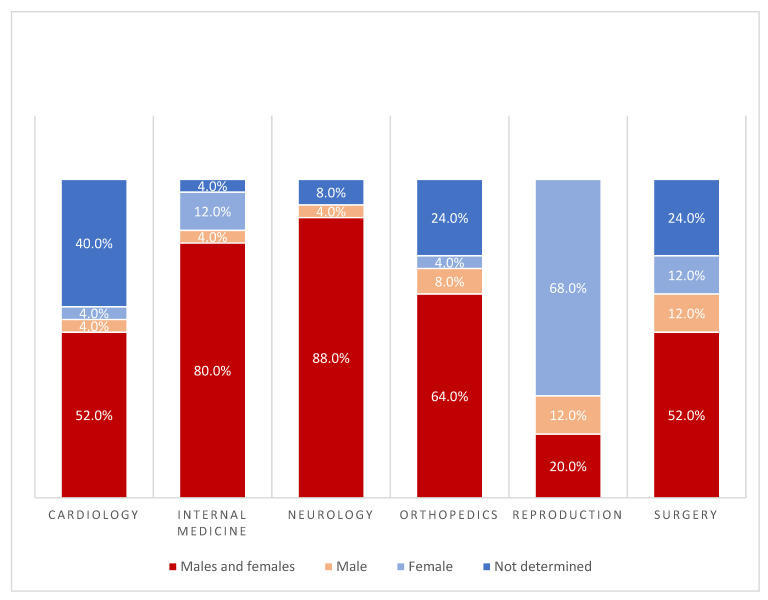
Sex of the dogs utilized in 150 clinical trials within six veterinary specialties (25 each).

**Figure 2 animals-12-01487-f002:**
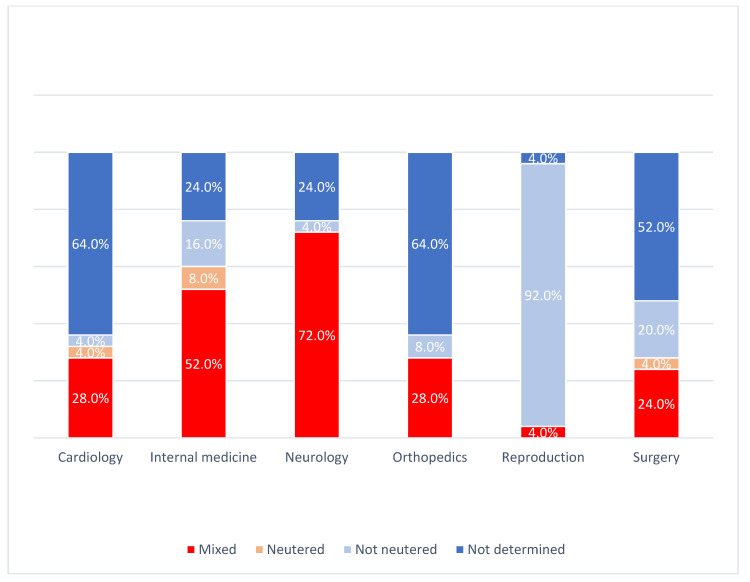
Spay and neuter status of the dogs utilized in 150 clinical trials within six veterinary specialties (25 each).

**Figure 3 animals-12-01487-f003:**
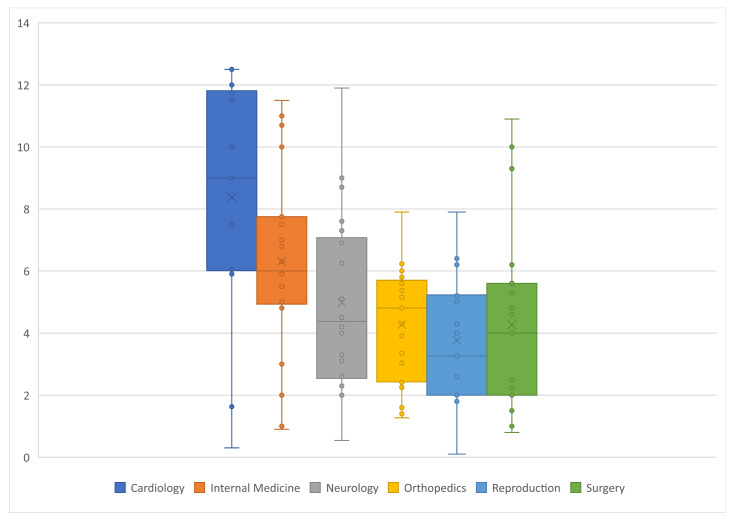
Age (in years) of the dogs utilized in 150 clinical trials within six veterinary specialties (25 each).

**Figure 4 animals-12-01487-f004:**
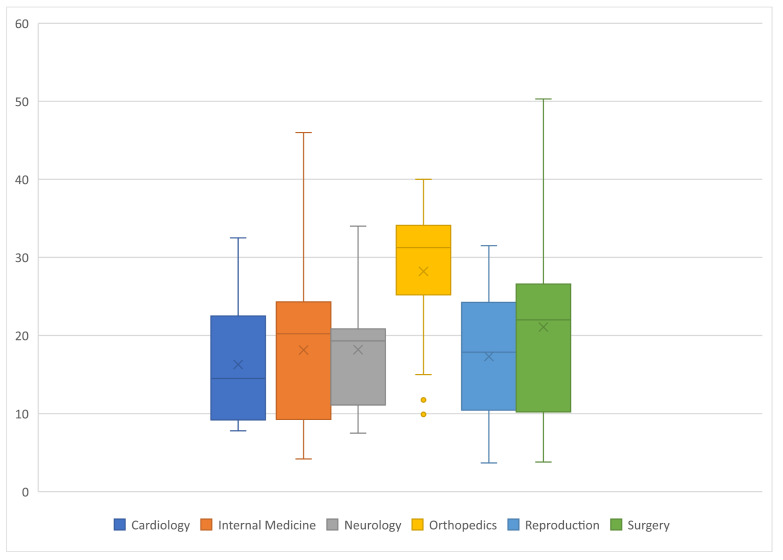
Weight (in kg) of the dogs utilized in 150 clinical trials within six veterinary specialties (25 each).

**Table 1 animals-12-01487-t001:** Top 10 dog breeds in the US in 2019 and the number of publications found in PubMed for each breed (search date: 21 May 2022).

Dog Breed	2019 Rank	Number of Publications Between 2007 and 2019 Resulting from the Breed Used as a Search Term in PubMed in May 2022
Labrador Retriever	1	824
German Shepherd	2	625
Golden Retriever	3	551
French Bulldog	4	115
Bulldog	5	488
Poodle	6	243
Beagle	7	5.199
Rottweiler	8	214
German Shorthaired Pointer	9	45
Pembroke Welsh Corgi	10	49

**Table 2 animals-12-01487-t002:** Use of Beagles as trial dogs within six veterinary specialties in 150 clinical trials (25 per specialty).

Specialty	Number of Studies (*n* = 25 per Specialty)	
	Using Beagles	With Missing Breed Information
Cardiology	4 (16.0%)	7 (28.0%)
Internal medicine	8 (32.0%)	10 (40.0%)
Neurology	5 (20.0%)	5 (20.0%)
Orthopaedics	4 (16.0%)	10 (40.0%)
Reproduction	3 (12.0%)	8 (32.0%)
Surgery	8 (32.0%)	6 (24.0%)
Total	32 (21.3%)	46 (30.7%)

**Table 3 animals-12-01487-t003:** Numbers and proportions of Beagle dogs utilized in clinical trials within six veterinary specialties (*n* = 150).

Specialty	Total Number of Dogs per Specialty (25 Studies)	Total Number of Dogs with Breed Specification	Proportion of Beagles in Relation to All Dogs with Known Breed
Cardiology	3319	850	4.4%
Internal medicine	24,696	286	12.9%
Neurology	2004	639	2.0%
Orthopaedics	433,347	784	5.6%
Reproduction	102,104	744	0.4%
Surgery	31,072	30,532	1.2%
Total	596,542	33,835	1.9%

**Table 4 animals-12-01487-t004:** Numbers and proportions of dogs belonging to the US top ten breeds of 2019 used in research (*n* = 150).

Dog Breed	Number (and Percentage) of Dogs of the US Top Ten Breeds of 2019 Used in 150 Clinical Trials
Labrador Retriever	2828 (6.7%)
German Shepherd	3055 (7.3%)
Golden Retriever	1780 (4.7%)
French Bulldog	51 (0.1%)
Bulldog	46 (0.1%)
Poodle	750 (1.8%)
Beagle	640 (1.5%)
Rottweiler	1228 (2.9)
German Shorthaired Pointer	78 (0.1%)
Pembroke Welsh Corgi	1 (0.002%)

## Data Availability

The data that support the findings of this study are available from the corresponding author upon reasonable request.

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
