# Peer review of "What Kinds of Dogs Are Used in Clinical and Experimental Research?"

_animals, 2022, doi:10.3390/ani12121487_

Round 1

Reviewer 1 Report

Review: What kind of dogs are used in veterinary research?

Dear authors,

Thank you very much for submitting this work – which I would like to see published – to Animals.

The topic is relevant and very interesting and I think it deserves further investigation.

I have some comments which I hope will help to improve the quality of this work.

I find that the title is misleading and should be reformulated, as it clearly refers to veterinary research; however, later in the text experimental human medicine studies are mentioned. Something like “use of dogs in clinical and experimental research” would be more appropriate.

I believe that a clearer distinction should be made between these two separate categories of research (clinical and experimental), and that the conclusions should also be drawn separately.

Moreover, within the category “veterinary studies”, there is a big difference between clinical and experimental studies. I am not surprised to hear that Beagle dogs are underrepresented in the first category; in fact, clinical studies are likely to focus on a certain specific condition and therefore the study population is likely to be standardised based on breed-predisposition. This explains why French Bulldogs (BOAS and upper airway abnormalities) and Labradors (OA/DJD) are overrepresented but I believe this being true for clinical studies rather than experimental trials. I am not sure that we could say that these dogs are “used in research” as this implies an experimental use, whereas dogs used in clinical studies suffer from a naturally occurring disease and are usually client-owned dogs.

Based on the studies included in this study, saying that Beagles are underrepresented is a generalization and an inaccurate statement. In fact, the authors report that Beagle are “used consistently” in human medicine experimental studies, which is expected…what would not be expected is for them to be highly represented in veterinary clinical studies investigating naturally occurring conditions.

I would like to see this paper taking into consideration these important differences, and making separate conclusions based on this.

With kind regards

Reviewer 2 Report

This is a well written and interesting paper showing that researchers need to be more aware of the reporting of their research findings in regard details of the animal subjects.  It is unclear to me exactly what the study populations are, and I am confused by the term clinical trial.  It seems to me the following areas need to be separated for analysis.

  1. Experimental Veterinary Research: Experimental studies in a research laboratory such as those under the Directive 2010/63/EU. They may be in any of the 6 specialised areas listed in the paper and will be carried out using purpose bred animals from a licensed breeder (e.g. mainly Beagles) or animals sourced from a licensed supplier (unowned stray animals – e.g. in the USA Category B dealers but this source is not permitted in the EU).

  1. Clinical Veterinary Research: Studies exempt from the Directive 2010/63/EU and carried out as part of non-experimental clinical veterinary practices under Article 1.5.b,c  of the Directive 2010/63/EU which are carried out in normal veterinary practice, outside a research laboratory and its controls, where the subjects could be companion dogs owner by members of the public and subject to a variety of environmental conditions as well as a variety of breeds, crossbred animals and reproductive capacity.

  1. Clinical Veterinary Trials: The definition of clinical trials needs to be explained and whether it is different or part of clinical veterinary research or experimental veterinary research. Clinical trials in human and veterinary medicine are a specific category of research often carried out to determine the effectiveness of a novel drug (medicine) before licensing (e.g. EU Directive cited above; Article 1.5.c), or novel dose or route of administration of a licensed medicine.  Some of these early trials (e.g. on ADME) would be under carefully controlled laboratory conditions and on standardised animal subjects.  Clinical veterinary research is far broader and would cover surgical approaches, devices and diagnostic tests as well as clinical drug trials, and be carried out using dogs belonging to general public which could be any of the breeds listed and include a range of cross-bred animals.

  1. The other imprecise groupings and generic words used include ‘medical research’ (is the purpose to benefit to humans or animals), testing (normally drug development not for medical devices or diagnosis).

  1. In the UK the Annual Statistics on experimental dog use in 2020. https://assets.publishing.service.gov.uk/government/uploads/system/uploads/attachment_data/file/1002895/annual-statistics-scientific-procedures-living-animals-2020.pdf

show that 99% of dogs used in experimental research (medical and veterinary) were Beagles and only other breeds.  In the EU the breed of dog is not specified in the statistics.  (https://ec.europa.eu/environment/chemicals/lab_animals/pdf/SWD_%20part_A_and_B.pdf

The authors use the word ‘attributes’, whereas the term most commonly used in clinical veterinary practice would be ‘signalment’. 

I suggest that the authors use these legal definitions more carefully in their paper, BUT the message they wish to convey is important and will still be the same.

Reviewer 3 Report

In the present work, authors analysed retrospectively information relative to the use of dogs in veterinary research concerning relevant data as breed, age, sex and reproductive status. Authors conclude that an important percentage studies lack some important information of the animals analysed and suggest improvements.   

The study is pertinent and answers to an important issue, as very little is known about the animals used in veterinary medicine studies.  Considering the actual volume and importance of veterinary research publications, this information is very important. A previous work has been published by the same authors concerning the same data, but with different objectives.

-General comments

Despite the interest and pertinence of the study, several aspects must be considered and reviewed before publication.

1-      One interesting information that would help to situate and clarify the context of the studies evaluated was to know which kind of studies did authors analysed. It would be interesting to know which studies were: experimental studies, as clinical trials; or observational studies (retrospective or prospective longitudinal -the “cohort” studies- and transversal studies). For observational studies, the dog population founded is very linked to specific region and/or country, which should always be looked with caution.

2-      On the other hand, authors confront, in Introduction section, the 2020 US breed dogs with the number of publications in Pubmed in March 2022. This does not seem adequate in terms of place in the manuscript, in terms of period analysed and in terms of geographic area which is being studied, as US top dog breeds are being compared to breeds of dogs in publications originated from all over the word in a very different temporal period. Additionally, the reference from which data of the top dog breeds were obtained is not given. Tables must be referred before they appear in the manuscript.

3-      The same concept is repeated in Table 4. Authors confront the US top breeds to breeds of the 150 studies published between 2007 to 2019. Besides, the caption in the Table refers to “150 clinical trials”; please confirm the term is correct. The insertion of this Table, as Table 1 must be reviewed, as they compare different periods and different topics.

4-      It will be very important that the manuscript could be revised by English native speaker/scientist for revision of phrase construction.

-Specific comments

 1- In the simple summary (L 8) the temporal period of the study could be referred in order to better situate the reader.

2- Please revise phrase construction at L 10 and L 11.

3- Also in Methods at L 28, the number of dogs could be given in order to give a more complete picture.

4- Please revise phrase construction at L 32 to L34

5-      Please revise phrases at L 73 to L 75, as well

6-      At L 102: the breed differences of incidence of pyometra will be very dependent of the canine demographic data of that specific area or country, as other works prove the contrary.

7-      In results section at L 142 to L 143, why to include a study in which the number of dogs is not specified? Wasn’t this an exclusion criterion? Please clarify. Besides, the reference of the study could be given.

8-      In Discussion section it could be better contextualised the perceptions around the use of the Beagle dogs. In one hand authors affirm that the proportion was less than expected (at L 266) then they declare latter (L 271) that there is a slight increase on the use of this breed. Please verify and adapt.

Round 2

Reviewer 1 Report

Dear authors,

Thank you for addressing my comments and concerns. I am happy with the answers you provided and with the changes you made to your original submission.

Kind regards

Reviewer 2 Report

The authors have addressed all of my major concerns.  It would be helpful if they mentioned the limitations of their research as they have done in the responses to the reviewers; comments.